# Comparative Mitochondrial Genomes between the Genera *Amiota* and *Phortica* (Diptera: Drosophilidae) with Evolutionary Insights into D-Loop Sequence Variability

**DOI:** 10.3390/genes14061240

**Published:** 2023-06-09

**Authors:** Caihong Zhang, Yalian Wang, Hongwei Chen, Jia Huang

**Affiliations:** 1Department of Entomology, South China Agricultural University, 483 Wushan-Lu, Guangzhou 510642, China; 2Guangdong Provincial Key Laboratory of Insect Developmental Biology and Applied Technology, Institute of Insect Science and Technology, School of Life Sciences, South China Normal University, Guangzhou 510631, China

**Keywords:** mitogenome, phylogeny, control region, Steganinae

## Abstract

To address the limited number of mitochondrial genomes (mitogenomes) in the subfamily Steganinae (Diptera: Drosophilidae), we assembled 12 complete mitogenomes for six representative species in the genus *Amiota* and six representative species in the genus *Phortica*. We performed a series of comparative and phylogenetic analyses for these 12 Steganinae mitogenomes, paying special attention to the commonalities and differences in the D-loop sequences. Primarily determined by the lengths of the D-loop regions, the sizes of the *Amiota* and *Phortica* mitogenomes ranged from 16,143–16,803 bp and 15,933–16,290 bp, respectively. Our results indicated that the sizes of genes and intergenic nucleotides (IGNs), codon usage and amino acid usage, compositional skewness levels, evolutionary rates of protein-coding genes (PCGs), and D-loop sequence variability all showed unambiguous genus-specific characteristics and provided novel insights into the evolutionary implications between and within *Amiota* and *Phortica*. Most of the consensus motifs were found downstream of the D-loop regions, and some of them showed distinct genus-specific patterns. In addition, the D-loop sequences were phylogenetically informative as the data sets of PCGs and/or rRNAs, especially within the genus *Phortica*.

## 1. Introduction

As the secondary genome in eukaryotes, the mitochondrial genome (mitogenome) is far from being independent of the nuclear genome and the rest of the cell [1]. The mitogenome encodes key parts of the oxidative phosphorylation complexes and is of vital importance for cellular fitness and organismal health [2,3]. Recently, we have gained an in-depth understanding of the mammalian mitochondrial transcriptional machinery and its relevance in cancer, inherited genetic disorders, inflammation, and neurodegenerative diseases [4,5]. The mitogenome and the nuclear genome adjust with respect to each other and develop corresponding genetic polymorphisms during the course of evolution [1,6]. Metazoan mitogenomes have been widely used for species identification, population genetics, comparative genomics, and phylogenetic studies in the past decade [7,8,9,10] due to their well-known advantages in availability and practicality [11,12].

In general, a typical mitogenome is a circular, double-stranded DNA molecule that is 14–20 kb long and contains 13 protein-coding genes (PCGs), 22 transfer RNA (tRNA) genes, two ribosomal RNA (rRNA) genes, and an A + T-rich, noncoding displacement loop (D-loop) region (also known as the control region) [7,13] relating to the control of mitochondrial DNA (mtDNA) gene transcription and mitochondrial replication [5,14]. As the most variable region in the mitogenome, the D-loop region contains the origin of replication for the majority strand (J-strand), and altered methylation and hydroxymethylation levels in the mammalian D-loop region have been associated with cancer, aging, metabolic disorders, cardiovascular diseases, and neurodegenerative diseases [15,16,17,18,19]. However, the genus-specific characteristics and evolutionary implications of D-loop region sequence variability have not been fully investigated and assessed.

The Drosophilinae and Steganinae subfamilies, which include more than 3500 and 1100 species worldwide, respectively [20], form the family Drosophilidae (pomace flies or small fruit flies). The Steganinae subfamily is diverse in its geographical distribution, morphology, and habit. *Amiota* Loew, 1862 [21] and *Phortica* Schiner (1862) [22] are currently the fourth and third largest genera in the Steganinae, respectively. Species in these two genera are mostly distributed in forests and share the common behavior of hovering in front of mammalian eyes or shiny objects. Some species are medically significant due to being the intermediate hosts of the Oriental eye worm, *Thelazia callipaeda* (Railliet & Henry) (Rhabditida: Thelaziidae), and therefore the epidemiologic factors for thelaziasis. *Phortica* was regarded as a subgenus of *Amiota* by Wheeler in 1952 [23] and remained that way for more than half a century until Máca recovered its generic rank in 2003 [24]. The most recent molecular phylogenetic studies confirmed the monophyly of the *Amiota* and the *Phortica* [25,26]. However, most of these studies were performed based on only one or three molecular genetic markers, which is not enough for the results to be considered convincing. In addition, the Steganinae is poorly studied in almost every field except taxonomy [27,28]. There is only one complete Steganinae mitogenome of *Phortica* (*Phortica*) *variegata* (Fallén) available in GenBank thus far, which is inadequate based on the extremely large number of taxa within this subfamily.

To address the limited number of Steganinae mitogenomes, we assembled six representative *Amiota* species and six representative *Phortica* species based on whole-genome sequencing data. These newly accessible mitogenomes enabled us to perform the first comparative analysis of the Steganinae, with an emphasis on the commonalities and differences between the *Amiota* and *Phortica* mitogenomes.

## 2. Materials and Methods

### 2.1. Specimen Collection and DNA Extraction

This study is based on thousands of Steganinae adult specimens collected at several sites in China over the last decade. We collected the specimens by net-sweeping around human eyes or along tree trunks in forests. All specimens were immediately preserved in 75% alcohol, transferred to −20 °C, and identified by the authors based on morphological characteristics. We selected the specimens of six representative *Amiota* species from five species groups (one ungrouped species) and five representative *Phorita* species belonging to three subgenera (Table 1) for further total DNA extraction, library construction, and whole-genome sequencing. The total DNA of each species was extracted from a complete individual using the TIANamp Genomic DNA Kit (#DP304, Tiangen Biotech., Beijing, China) according to the manufacturer’s protocol.

### 2.2. Library Construction and Sequencing

Berry Genomics (Beijing, China) carried out the construction of whole-genome sequencing libraries and the subsequent sequencing. Briefly, the total DNA samples were sonicated and split into random fragments by a Bioruptor^®^ Pico (Diagenode, Belgium), and fragments of ~350 bp were selected for library construction using the Illumina^®^ DNA Prep, (M) Tagmentation kit (#20060059, Illumina, CA, USA) for adaptor ligation and PCR amplification with index sequences. The 11 constructed libraries were loaded and sequenced with a 2 × 150 bp paired-end (PE150) run on a NovaSeq 6000 platform (Illumina) after quality control.

### 2.3. Mitogenome Assembly and Annotation

Including the DNA sequencing data of a Nearctic *Phortica* species, *P*. (*P*.) *variegata*, available from the Sequence Read Archive (SRA) (accession number: SRR826812) (Table 1), we assembled or reassembled a total of 12 mitogenomes (six *Amiota* + six *Phortica* species) based on the reads from the raw sequencing data that contained mitogenome fragments. In our case, approximately 1–2% of reads were mitogenome-derived, which is sufficient to assemble a complete mitogenome in most cases. Based on the overlap-layout-consensus (OLC) approach, we manually assembled each mitogenome using Geneious Prime v2020.0.5 [29], with multiple mapping iterations with suitable mismatch rates to either a reference mitogenome (the first mapping) or the gradually lengthening mitochondrial sequence itself (the other mapping iterations). A complete mitogenome was obtained once the two ends of the sequence highly overlapped with each other and could no longer be lengthened. All mitogenome sequences were subjected to Basic Local Alignment Search Tool (BLAST) [30] comparisons to ensure the correct taxonomic identification results.

We primarily ascertained the rough boundary of each mitochondrial gene using the MITOS2 Web Server [31]. Accurate boundaries of PCGs and rRNAs were determined by aligning with the reference mitogenome of *D*. *melanogaster* (NC024511) using MEGA v11.0.13 [32]. Locations of mitochondrial tRNAs were identified using the tRNAscan-SE Search Server v2.0 [33].

### 2.4. Mitogenome Analyses

We regarded *A*. *femorata* and *P*. (*P*.) *variegata* as the representative species of each genus and depicted the graphical maps of these two mitogenomes using the Proksee server [34]. Nucleotide composition and codon usage of the PCGs were calculated in MEGA. Compositional skewness levels were estimated in terms of the following formulas: AT-skew = (A − T)/(A + T) and GC-skew = (G − C)/(G + C) [35]. We performed PCAs based on the gene (Appendix A) sizes, the IGN sizes (Appendix A), the RSCU values (Appendix A), and the CDspT values (Appendix A) of the 12 mitogenomes using the built-in R stats package v4.0.3 [36]. The RSCU values were displayed in a heatmap generated by the R package pheatmap v1.0.12 [37]. The rates of Ka and Ks of each PCG within the *Amiota* or *Phortica* were calculated by the “Compute Pairwise Distances” module in MEGA and visualized using the R package ggplot2 v3.3.3 [38]. We performed de novo motif discovery on the 12 D-loop sequences using the “MEME” module in MEME Suite v5.4.1 [39] with the main parameters “-dna -mod zoops-nmotifs 20-minw 10-maxw 50-objfun classic-markov_order 0”.

### 2.5. Phylogenetic Analyses

To rigorously test the monophyly of Steganinae, a total of 14 complete drosophilid mitogenomes were included in subsequent phylogenetic analyses, including the 12 assembled Steganinae mitogenomes and two Drosophilinae mitogenomes [*D*. *melanogaster* (NC024511) and *Zaprionus indianus* Gupta (MK659852)] as outgroup taxa. We performed phylogenetic analyses on the following three data sets of mitogenomes: (1) nucleotides of concatenated 13 PCGs and two rRNAs; (2) amino acids of concatenated 13 PCGs; and (3) nucleotides of the D-loop regions. The three data sets were aligned using MAFFT v7.490 [40] with the L-INS-I algorithm (––localpair ––maxiterate 1000). For the concatenated data sets, we used PartitionFinder v2.1 [41] to search the best-fit partitioning schemes and substitution models based on a “greedy” algorithm and the corrected Akaike information criterion (AIC) scores. Nucleotides of the PCGs were partitioned by codon positions, whereas rRNAs and amino acids of the PCGs were partitioned by genes. We performed the BI using MrBayes v3.2.6 [42], which ran two independent sets of Markov chains, each with one cold and three heated chains for a total of 10 million generations, sampling every 1000 generations. Chain convergence was measured by the standard deviation of split frequencies < 0.01 in MrBayes and the effective sample sizes (ESSs) ≥ 200 in Tracer v1.7.1 [43]. The consensus tree from each BI run was generated after 25% of the trees had been discarded as a burn-in. ML analyses were conducted using IQ-TREE v1.6.12 [44,45]. UBPs for the majority-rule consensus tree in each ML analysis were calculated with 1000 replicates [46]. All resulting trees were visualized using FigTree v1.4.3 [47] with adjustable settings.

## 3. Results and Discussion

### 3.1. Mitogenome Organizations and Base Compositions

In this study, we assembled 12 mitogenomes first for the Steganinae, including *Amiota dentata* Okada, *Amiota femorata* Chen & Takamori, *Amiota nagatai* Okada, *Amiota setosa* Zhang & Chen, *Amiota spinifemorata* Li & Chen, *Amiota yifengi* Zhang & Chen, *Phortica* (*Ashima*) *longipenis* Chen & Gao, *Phortica* (*Ashima*) *tanabei* Chen & Toda, *Phortica* (*Phortica*) *huazhii* Cheng & Chen, *Phortica* (*Phortica*) *pseudogigas* Zhang & Gan, *P*. (*P*.) *variegata*, and *Phortica* (*Shangrila*) *hani* Zhang & Shi (Table 1). They were all closed-circular, double-stranded DNA molecules, and therefore complete mitogenomes. Represented by the representative *Amiota* (*A*. *femorata*) and *Portica* species (*P*. (*P*.) *variegata*), all 12 Steganinae mitogenomes contained the typical set of 37 genes (13 PCGs, 22 tRNAs, and two rRNAs) and had an expected D-loop region located between srRNA and *trnI* (Figure 1), which was consistent with most metazoan mitogenomes [8,11,12]. Nine of the 13 PCGs and 14 of the 22 tRNAs were encoded by the J-strand, and the remaining 12 genes were encoded by the minority strand (N-strand) (Figure 1 and Table 2). In addition, the gene arrangements of these 12 Steganinae mitogenomes were highly conserved and identical to that of *Drosophila* (*Sophophora*) *yakuba* Burla (Figure 1 and Table 2), which is one of the most common gene arrangement types among arthropod mitogenomes [48].

The lengths of the six *Amiota* mitogenomes ranged from 16,143 bp in *A*. *spinifemorata* to 16,803 bp in *A*. *setosa*, whereas the six *Phortica* mitogenomes ranged from 15,933 bp in *P*. (*A*.) *tanabei* to 16,290 bp in *P*. (*A*.) *longipenis* (Table 1 and Table 2). All of them were within the size range of known complete Drosophilinae mitogenomes from 14,874 bp in *Drosophila* (Hawaiian *Drosophila*) *grimshawi* Oldenberg (BK006341) to 19,951 bp in *Drosophila* (*Sophophora*) *melanogaster* Meigen (CM010568). In the principal component analysis (PCA) based on the sizes of genes, PC1 and PC2 explained 45.94% of the total variance (Figure 2A). The *Amiota* species could be separated from the *Phortica* species. Furthermore, the three subgenera of the *Phortica* also showed distinct patterns at the subgenus level.

The variation of gene intergenic nucleotides (IGNs) is another important characteristic of the mitogenome. For instance, both the *A*. *femorata* and *P*. (*P*.) *variegata* mitogenomes contained a total of 121 bp gene overlaps or IGNs ranging from −1 to 18 bp, whereas they were located in 21 and 19 pairs of adjacent genes, respectively (Table 2). The longest IGNs of 18 bp in both the *A*. *femorata* and *P*. (*P*.) *variegata* mitogenomes were located between *trnE* and *trnF*. However, we found another 18 bp IGNs between *trnS1* and *ND1* in the latter (Table 2). According to the PCA based on the sizes of IGNs, PC1 and PC2 explained 50.17% of the total variance (Figure 2B). The *Amiota* species also remained some distance away from the *Phortica* species. This PCA showed the specific size patterns of IGNs within the *Amiota* and *Phortica* mitogenomes. In addition, *A*. *nagatai* had four pairs of adjacent tRNAs with 32–49 bp IGNs, and *P*. (*A*.) *longipenis* had 100 bp IGNs between *trnI* and *trnQ*. The generation of these long IGNs between tRNAs might be related to the evolutionary divergence of these species.

The base compositions of the 12 Steganinae mitogenomes were A = 39.1–40.7%, T = 38.1–40.3%, G = 8.1–9.5%, and C = 11.2–13.6% (Appendix A). Among the whole mitogenome, PCGs (and each codon position), tRNAs, rRNAs, and D-loop regions, the *Amiota* mitogenomes generally had a higher A + T content, whereas the *Phortica* mitogenomes generally had a higher G + C content (Appendix A). The J-strands and the D-loop regions generally had positive AT-skews and negative GC-skews with the exception of *A*. *nagatai*, indicating no reversal of strand asymmetry in these Steganinae mitogenomes. Both the PCGs and the 1st codon positions had negative AT- and GC-skews, but all the 2nd codon positions had negative AT-skews and positive GC-skews (Appendix A and Figure 3). By establishing the linear relationships between AT-skew and GC-skew in the *Amiota* and *Phortica* mitogenomes (Appendix A and Figure 3), we found the following: (1) the AT- and GC-skews were relatively correlated only for the J-strands (R^2^ = 0.519–0.884, *p* = 0.005–0.106) (Figure 3A); (2) the linear relationships between the *Amiota* and *Phortica* were relatively less overlapped in the 2nd (Figure 3D) and 3rd codon positions (Figure 3E), tRNAs (Figure 3G), and D-loop regions (Figure 3H); and (3) the compositional skewness levels in the 1st codon positions (Figure 3C) and rRNAs (Figure 3G) were relatively specific for the *Amiota* and the *Phortica* (reflected by less overlap). Altogether, our results showed unambiguous genus-specific characteristics between and within the *Amiota* and *Phortica* from mitochondrial base composition to skewness.

### 3.2. Protein-Coding Genes

Accounting for 66.4–70.2% of the entire Steganinae mitogenome, the lengths of the 13 PCGs in the *Amiota* and *Phortica* ranged from 11,152–11,178 and 11,182–11,184 bp, respectively (Appendix A). The initial codons for the 13 PCGs of the *A*. *femorata* and *P*. (*P*.) *variegata* mitogenomes were putative start codons ATD, except for *COX1*, which used TCG (Table 2) as in other dipteran insects [11,14,49,50,51]. *ATP8* and *ND1* severally started with ATT and ATA in the *A*. *femorata* mitogenome, whereas they started with ATC and ATT in the *P*. (*P*.) *variegata* mitogenome (Table 2).

Ten PCGs in the *A*. *femorata* mitogenome shared the termination codon TAR, whereas the remaining *ND1*, *ND5*, and *ND6* terminated with an incomplete stop codon T (Table 2). Similar to the *A*. *femorata* mitogenome, the *P*. (*P*.) *variegata* mitogenome shared identical termination codons excluding *COX2*, which also used an incomplete stop codon T (Table 2). A common interpretation for the incomplete stop codon T is that TAA termini can be produced by posttranscriptional polyadenylation [52].

Apart from the stop codons, a total of 3707–3717 amino acids in the Steganinae mitogenomes were encoded by the mitochondrial PCGs. In the heatmap based on the relative synonymous codon usage (RSCU) values of the PCGs, the *Amiota* and *Phortica* species separated from each other in the cluster analysis (Figure 4A). The use of both two- and four-fold degenerate codons revealed a bias toward nucleotides A and T, especially in the 3rd codon positions (90.3–96.5%) (Appendix A). Overall, the seven most frequently used codons were UUA (Leu1), UCU (Ser1), CGA (Arg), CCU (Pro), GGA (Gly), GCU (Ala), and UCA (Ser1), which constituted 32.5–34.6% of the total amino acid codons in these mitogenomes (Appendix A). Including AGG (Ser2), two to six of the 62 possible degenerate codons were absent (Figure 4A). The loss of degenerate codons probably randomly occurred during evolution [53]. The two *Ashima* species, *P*. (*A*.) *longipenis* and *P*. (*A*.) *tanabei*, were both missing three degenerate codons. Only the codon family of Leu1 had codons per thousand codons (CDspT) values over 100 in all of the Steganinae mitogenomes (Appendix A). As a hydrophobic amino acid, Leu1 might be associated with the encoding of transmembrane proteins [54]. The codon family with the second highest CDspT values, Ile, was more abundant in the *Amiota* mitogenomes (100.51–106.17) than in the *Phortica* mitogenomes (93.97–99.89) (Appendix A). PC1 and PC2 explained 48.55% and 64.29% of the total variance to the PCAs based on the RSCU (Figure 4B) and CDspT values (Figure 4C), respectively. Similar to the characteristics of gene sizes (Figure 2A), the *Phortica* species separated from the *Amiota* species with distinct patterns at the subgenus level (Figure 4B,C). Furthermore, the pattern of the CDspT values was more specific than that of the RSCU values (Figure 4B,C). All these results revealed unambiguous population characteristics and provided novel insights into the evolutionary implications between and within the *Amiota* and *Phortica* based on the codon and amino acid usage of the mitochondrial PCGs.

As the indicators of evolutionary relationships and selective pressures of different species, the nonsynonymous substitutions per nonsynonymous site (Ka), synonymous substitutions per synonymous site (Ks), and Ka/Ks (*ω*) values of the 13 PCGs were computed for the *Amiota* and *Phortica* mitogenomes (Appendix A). In this study, the Ka values of 12 PCGs in the six *Amiota* mitogenomes were significantly higher than those in the six *Phortica* mitogenomes, whereas eight PCGs had significantly higher Ks values in the *Phortica* mitogenomes than in the *Amiota* mitogenomes (Figure 5). The generally opposite Ka and Ks values resulted in consistently and significantly higher *ω* values in all the PCGs of the *Amiota* mitogenomes compared with the *Phortica* mitogenomes (Figure 5 and Appendix A). In addition, our results showed that the significance of the *ω* values (0.001 < *p* < 0.01) of the ATPase subunits (*ATP6* and *ATP8*) was inferior to those of other mitochondrial PCGs (*p* < 0.001) (Figure 5). All *ω* values < 1 (Ka < Ks) indicated that the PCGs evolved under purifying selection [55]. Our results suggested that the PCGs with relatively low evolutionary rates, including *COX1*, *CYTB*, and *NAD1* (Figure 5), are better mitochondrial genetic markers than other PCGs for phylogenetic studies at both the nucleotide and amino acid levels.

### 3.3. Transfer RNA and Ribosomal RNA Genes

The typical set of 22 tRNA genes was scattered throughout each circular, double-stranded DNA molecule ranging from 62–64 bp (*trnC*) to 72 bp (*trnV*) (Appendix A). The sizes of the total tRNAs in the *Amiota* and *Phortica* mitogenomes ranged from 1440–1449 bp and 1434–1440 bp, respectively (Appendix A). All the tRNA genes had typical cloverleaf secondary structures apart from *trnSer1*, which lost the dihydrouracil (DHU) stem but held the DHU loop. Although *trnSer1* in some other dipteran mitogenomes also did not form complete cloverleaf secondary structures, they generally had neither the DHU stem nor the DHU loop [11,14,56].

The two mitochondrial rRNA genes that were encoded by the N-strand were located between *trnL2* and *trnV* (lrRNA) and between *trnV* and the D-loop region (srRNA), respectively (Figure 1 and Table 2). In the *Amiota* mitogenomes, lrRNA was 1322–1338 bp in length, whereas the lrRNA sizes in the *Phortica* mitogenomes ranged from 1317–1323 bp (Appendix A). The sizes of srRNA were 784–789 bp in the 12 Steganinae mitogenomes without a significant difference in length between the *Amiota* and *Phortica* mitogenomes (Appendix A).

### 3.4. The D-Loop Regions

The D-loop regions (J-strand) of the 12 Steganinae mitogenomes were all located between srRNA and *trnI* with variable lengths. The *Amiota* and *Phortica* D-loop regions were 1515–1995 bp with 93.0–94.0% A + T contents and 1084–1355 bp with 90.2–92.9% A + T contents, respectively (Appendix A). Furthermore, the size differences in the D-loop regions were crucial to the size differences in the mitogenomes. Our PCA results revealed that variability in the D-loop region was one of the important distinctions between the *Amiota* and *Phortica* mitogenomes, and even within the subgenus *Phortica* mitogenomes (Figure 2A).

In the de novo motif discovery analysis, we searched a total of 20 consensus motifs ranging from 10–50 bp in the 12 D-loop sequences (Figure 6). It is worth noting that most of the consensus motifs were distributed downstream next to *trnI* in the D-loop sequences. The upstream regions 200–600 bp away from srRNA had only two or three consensus motifs (Figure 6). The midstream regions between the abovementioned motif-enriched regions were highly variable in length and contained almost no consensus motifs (Figure 6). Given the positive correlation between the sizes and the A + T contents of the D-loop regions, it is clear that the nonconserved regions were mainly composed of tandem repeats with very high A + T contents [57].

Moreover, we identified some consensus motifs that showed distinct genus-specific patterns. For example, a “TWTWWYTMTHWAATAWWTAWYWTWWWTWHWHMTATATWTATWTAYADRH” consensus motif (motif 10, the yellow-green box in Figure 6) located in the downstream region approximately 300 bp away from *trnI* was found only in the *Phortica* D-loop sequences. Based on the above results, we are convinced of the importance of the genus-specific characteristics and evolutionary implications provided within the D-loop regions of the *Amiota* and *Phortica* mitogenomes. Although the D-loop regions were highly variable, their downstream conserved regions could be partly aligned to reflect the phylogenetic positions that were altered by evolutionary driving forces throughout evolution.

### 3.5. Phylogenetic Analyses

In this study, the phylogenetic relationships of three mitochondrial data sets of the 12 Steganinae and two Drosophilinae species were recovered (Figure 7). For the data set based on the nucleotides of 13 PCGs and two rRNAs and the data set based on the amino acids of 13 PCGs, the concatenated sequences were divided into 20 and six subsets using the best-fit substitution models, respectively (Appendix A). These two large data sets accounted for most of the genetic information in the mitogenomes. Overall, both the final Bayesian inference (BI) and maximum likelihood (ML) trees of these two data sets displayed an entirely identical topology, and most nodes were strongly supported (posterior probabilities, PPs = 1.00, ultrafast bootstrap percentages, UBPs = 100) (Figure 7A). The monophylies of the Steganinae, the Drosophilinae, the *Phortica*, and the *Amiota* were all well supported (PPs = 1.00, UBPs = 100). The phylogenetic relationships within the *Phortica* were also attached with high confidence (PPs = 1.00, UBPs = 100) with the exception between *P*. (*P*.) *huazhii* and *P*. (*P*.) *variegata* in the ML analyses (UBP = 99 for the nucleotides, UBP = 86 for the amino acids) (Figure 7A). The subgenera *Phortica* and *Shangrila* showed a closer relationship compared with the *Ashima*, which is consistent with the latest phylogenetic analysis concerning the *Phortica* [26]. *A*. *nagatai* (the *nagatai* species group) was primarily separated from the clade of the remaining five *Amiota* species (PP = 1.00, UBP = 100 for the nucleotides, PP = 0.91, UBP = 90 for the amino acids) (Figure 7A). The topology within this clade was supported in the BI (PPs = 0.60–1.00) but not well supported in the ML analyses (UBPs = 38–50). Nevertheless, all four trees reached a consensus on the close relationships between *A*. *setosa* (the *alboguttata* species group) and *A*. *femorata* + *A*. *spinifemorata* (the *taurusata* species group), and between *A*. *yifengi* (the *basdeni* species group) and *A*. *dentata* (ungrouped species) (Figure 7A).

The D-loop sequences were used as another data set for phylogenetic analyses. Briefly, apart from the phylogenetic transposition between *A*. *nagatai* and *A*. *setosa*, the topology of all other nodes was the same as in the above analyses (Figure 7). The nodal support values of the D-loop sequence data set were generally lower than those of the PCG data sets. This is expected since the D-loop sequences were highly variable and difficult to align due to the shortage of pairwise sites that could be used for phylogenetic calculations. However, the D-loop sequences still strongly supported the Steganinae, the Drosophilinae, the genus *Phortica*, the subgenus *Phortica*, and the *Amiota taurusata* species group each forming a monophyletic clade in both the BI and ML analyses (PPs = 0.99–1.00, UBPs = 95–100) (Figure 7B). In addition, the close relationship between *P*. (*P*.) *huazhii* and *P*. (*P*.) *variegata* was highly supported by the D-loop sequences (PP = 1.00, UBP = 100) (Figure 7B). The consistency in the phylogenetic topology of most clades indicated that the D-loop regions generally evolved under similar evolutionary pressures as the mitochondrial PCGs or the mitogenomes, whereas the inconsistency of partial phylogenetic topology implied that the D-loop motifs in the *Phortica* mitogenomes were more conservative and phylogenetically informative than those in the *Amiota* mitogenomes (Figure 6).

The D-loop region is inferred to play important roles in mtDNA gene transcription and mitochondrial replication [5,14]. Further efforts should be made to identify consensus motifs using gene-editing techniques such as CRISPR-Cas9 [58] to verify related regulatory mechanisms in the D-loop region. The rapid rate of evolution might make the D-loop region a promising molecular genetic marker to reveal the cause of the divergence of closely related species or subspecies [59,60]. In addition, we found that the genetic divergences within the *Amiota* were significantly greater than those within the *Phortica*. Therefore, we suggest further studies to investigate the necessity, possibility, and feasibility of establishing subgenera within the *Amiota*.

## 4. Conclusions

A total of 12 novel mitogenomes belonging to two large genera in the Steganinae were sequenced, assembled, annotated, compared, and analyzed in the present study. Our mitogenomic results strongly supported the Steganinae, the *Amiota*, the genus *Phortica*, the subgenera *Phortica*, and the *Shangrila* as a monophyletic group, respectively. The downstream D-loop sequences were also found to be phylogenetically informative and provided a similar topology to the Steganinae as the non-D-loop mitochondrial regions. Overall, these results not only broaden our knowledge of the mitochondrial characteristics of the Steganinae but also improve our understanding of the phylogenetic relationships within the subfamily. In addition, the D-loop sequence variability within these mitogenomes showed unambiguous genus-specific characteristics and provided novel insights into the evolutionary implications between and within the *Amiota* and *Phortica*. These newly available mitogenomes will contribute to further species identification, evolutionary biology, and conservation biology, and help to reveal the phylogenetic relationships and evolutionary history of the Drosophilidae.

## Figures and Tables

**Figure 1 genes-14-01240-f001:**
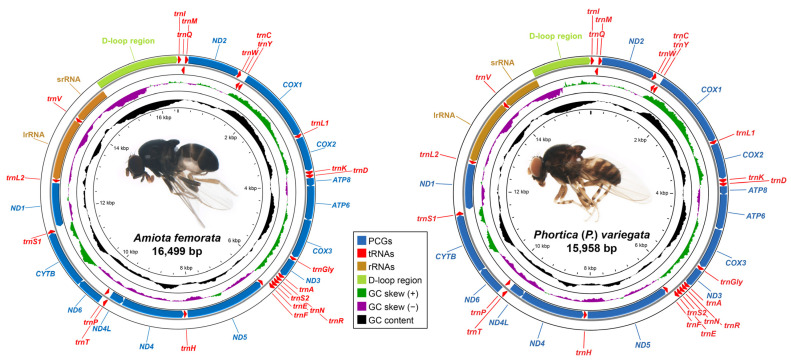
Graphical maps of the mitochondrial genomes (mitogenomes) of the representative *Amiota* (*Amiota femorata* Chen & Takamori) and *Phortica* species [*Phortica* (*Phortica*) *variegata* (Fallén)], respectively. The genes outside the outermost circle are transcribed clockwise, whereas the genes inside the outermost circle are transcribed counterclockwise. The inside circles show the G + C content and the GC-skews.

**Figure 2 genes-14-01240-f002:**
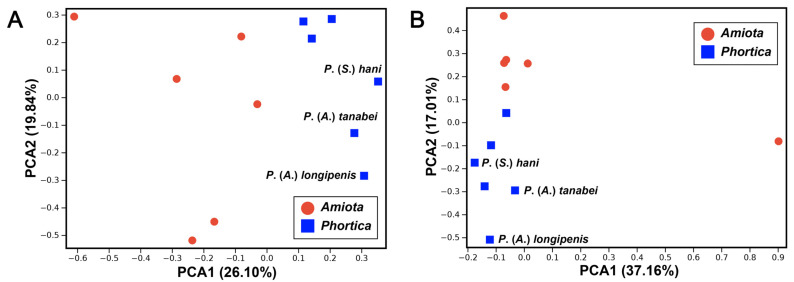
Principal component analyses (PCAs) based on (**A**) the sizes of genes and (**B**) the sizes of intergenic nucleotides (IGNs) in the *Amiota* and *Phortica* mitogenomes.

**Figure 3 genes-14-01240-f003:**
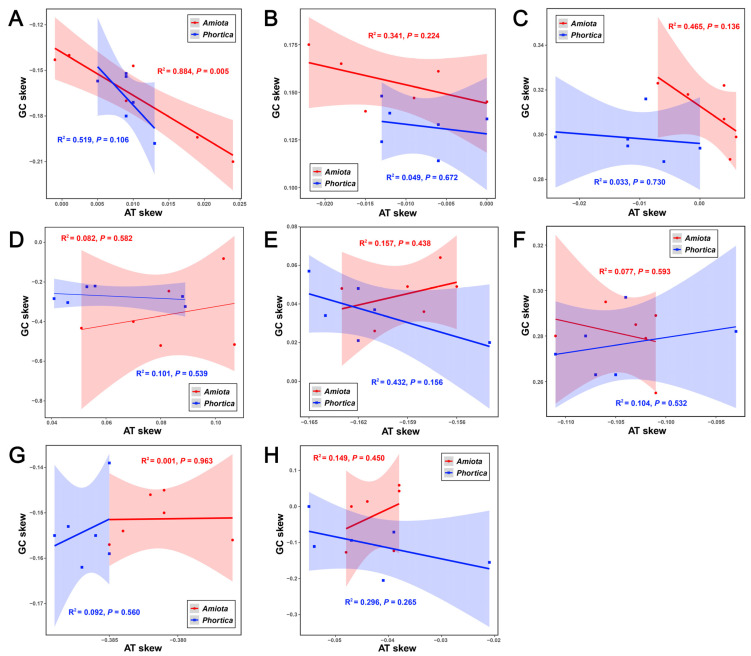
Linear correlation analyses between the AT-skews and the GC-skews of different regions/genes in the *Amiota* and *Phortica* mitogenomes. (**A**) The majority strands (J-strands) of whole mitogenomes. (**B**) Thirteen protein-coding genes (PCGs). (**C**) Thirteen PCGs (1st codon positions). (**D**) Thirteen PCGs (2nd codon positions). (**E**) Thirteen PCGs (3rd codon positions). (**F**) Twenty-two tRNAs. (**G**) Two rRNAs. (**H**) The D-loop regions (J-strands).

**Figure 4 genes-14-01240-f004:**
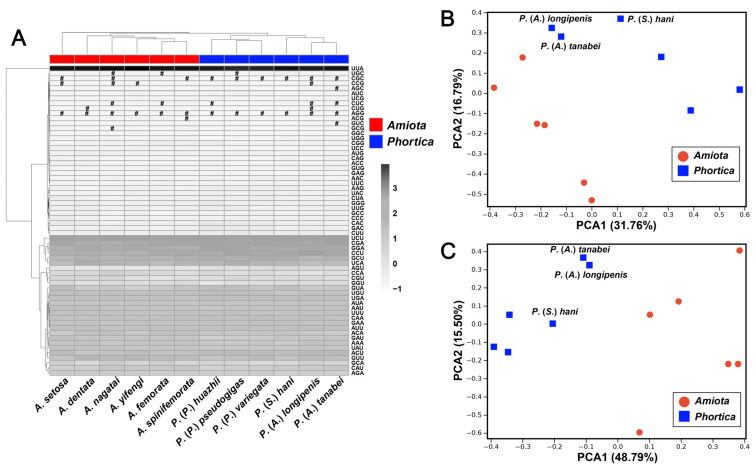
Genetic characteristics of the codon and the amino acid usage of all the PCGs in the *Amiota* and *Phortica* mitogenomes. (**A**) Heatmap and clustering analysis of the codon usage. Pound (#): absent codon. (**B**) PCA based on the relative synonymous codon usage (RSCU) values. (**C**) PCA based on the codons per thousand codons (CDspT) values.

**Figure 5 genes-14-01240-f005:**
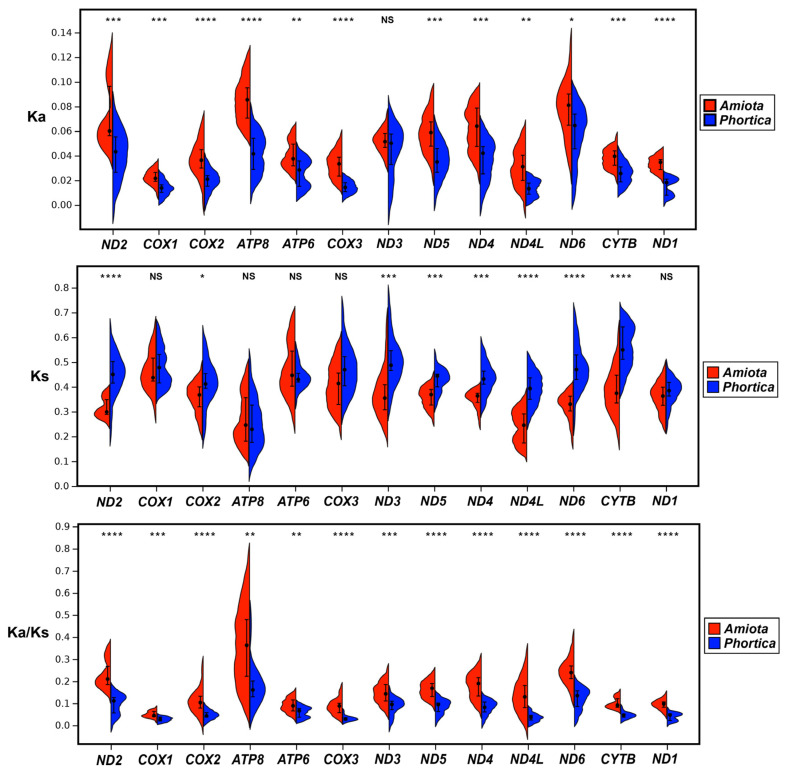
Nonsynonymous substitutions per nonsynonymous site (Ka), synonymous substitutions per synonymous site (Ks), and Ka/Ks (*ω*) values of all the PCGs in the *Amiota* and *Phortica* mitogenomes. In all analyses, (*) *p* < 0.05, (**) *p* < 0.01, (***) *p* < 0.001, (****) *p* < 0.0001, (NS) not significant.

**Figure 6 genes-14-01240-f006:**
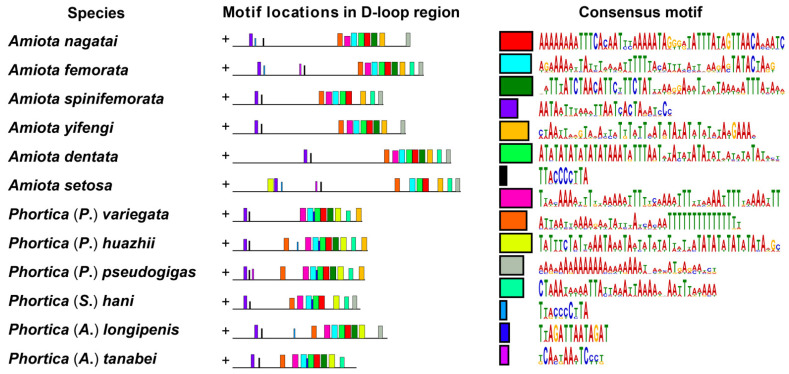
De novo motif discovery of the D-loop sequences (J-strands) in the *Amiota* and *Phortica* mitogenomes.

**Figure 7 genes-14-01240-f007:**
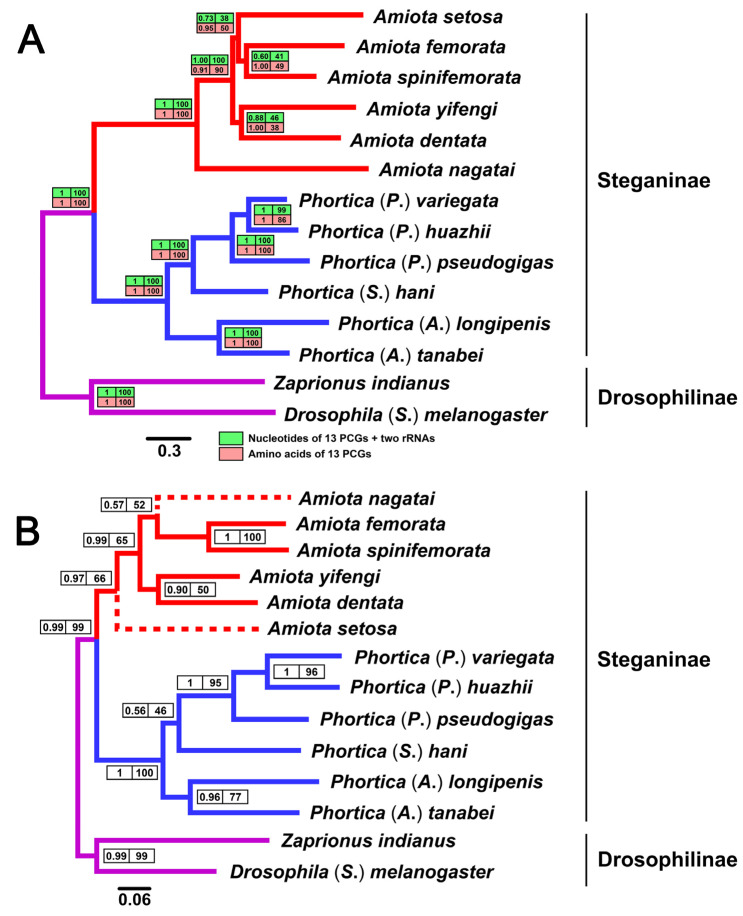
Phylogenetic trees reconstructed by Bayesian inference (BI) and maximum likelihood (ML) analyses for the data sets of 14 Drosophilidae mitogenomes. (**A**) Entirely identical topology inferred from the nucleotides of the 13 PCGs and two rRNAs (green in the upper boxes) and the amino acids of the 13 PCGs (pink in the lower boxes). (**B**) Similar topology to (**A**) inferred from the D-loop sequences. The dotted lines indicate that the *Amiota* species and clades are inconsistent with the topology displayed in (**A**). For both trees, numbers around nodes are posterior probabilities (PPs, left) calculated from the BI and ultrafast bootstrap percentages (UBPs) calculated from the ML analyses. The bar indicates the estimated number of substitutions per site.

**Table 1 genes-14-01240-t001:** Details of the *Amiota* and *Phortica* species and mitogenomes used in this study.

Genus	Subgenus/Species Group	Species	Collection Site	Mitogenome Accession Number	SRA Accession Number
*Amiota*	/	*dentata* Okada	Fengtongzhai, Baoxing, Sichuan, China	OP381033	SRR21438455
*Amiota*	*taurusata*	*femorata* Chen & Takamori	Dafengding, Mabian, Sichuan, China	OP381034	SRR21438454
*Amiota*	*nagatai*	*nagatai* Okada	Conghua, Guangzhou, Guangdong, China	OP381035	SRR21438452
*Amiota*	*alboguttata*	*setosa* Zhang & Chen	Dafengding, Mabian, Sichuan, China	OP381036	SRR21438451
*Amiota*	*taurusata*	*spinifemorata* Li & Chen	Gexigou, Yajiang, Sichuan, China	OP381037	SRR21438450
*Amiota*	*basdeni*	*yifengi* Zhang & Chen	Gexigou, Yajiang, Sichuan, China	OP381038	SRR21438449
*Phortica*	*Ashima*	*longipenis* Chen & Gao	Hesong, Menghai, Yunnan, China	OP381039	SRR21438448
*Phortica*	*Ashima*	*tanabei* Chen & Toda	Muyiji Park, Ximeng, Yunnan, China	OP381040	SRR21438447
*Phortica*	*Phortica*	*huazhii* Cheng & Chen	Ruili Park, Yunnan, China	OP381041	SRR21438446
*Phortica*	*Phortica*	*pseudogigas* Zhang & Gan	Mangshi, Yunnan, China	OP381042	SRR21438445
*Phortica*	*Phortica*	*variegata* (Fallén)	/	OP381043	^a^ SRR826812
*Phortica*	*Shangrila*	*hani* Zhang & Shi	Gexigou, Yajiang, Sichuan, China	OP381044	SRR21438453

Slash (/): not applicable or unknown. ^a^ The *Phortica* (*Phortica*) *variegata* (Fallén) mitogenome was assembled from the DNA sequencing data deposited in the NCBI Short Read Archive (SRA, https://www.ncbi.nlm.nih.gov/sra) (accessed on 6 September 2022).

**Table 2 genes-14-01240-t002:** Annotations and gene organizations of the mitogenomes of *Amiota femorata* Chen & Takamori and *Phortica* (*Phortica*) *variegata* (Fallén).

Gene	Strand	*A*. *femorata*				*P*. (*P*.) *variegata*			
		Position (bp)	Size (bp)	Anti- or Start/Stop Codons	IGN(s) (bp)	Position (bp)	Size (bp)	Anti- or Start/Stop Codons	IGN(s) (bp)
*trnI* (Ile)	+	1–65	65	GAU	7	1–65	65	GAU	15
*trnQ* (Gln)	−	73–141	69	UUG	−1	81–149	69	UUG	−1
*trnM* (Met)	+	141–209	69	CAU	0	149–217	69	CAU	0
*ND2*	+	210–1232	1023	ATT/TAA	0	218–1243	1026	ATT/TAA	0
*trnW* (Trp)	+	1231–1298	68	UCA	6	1242–1309	68	UCA	6
*trnC* (Cys)	−	1291–1353	63	GCA	2	1302–1364	63	GCA	2
*trnY* (Tyr)	−	1356–1421	66	GUA	0	1367–1432	66	GUA	0
*COX1*	+	1420–2958	1539	TCG/TAA	3	1431–2969	1539	TCG/TAA	3
*trnL1* (Leu) (UUR)	+	2954–3019	66	UAA	3	2965–3030	66	UAA	2
*COX2*	+	3023–3703	681	ATG/TAA	7	3033–3717	685	ATG/T-	3
*trnK* (Lys)	+	3711–3781	71	CUU	−1	3721–3791	71	CUU	0
*trnD* (Asp)	+	3781–3850	70	GUC	0	3792–3858	67	GUC	0
*ATP8*	+	3851–4012	162	ATT/TAA	5	3859–4020	162	ATC/TAA	5
*ATP6*	+	4006–4683	678	ATG/TAA	−1	4014–4691	678	ATG/TAA	6
*COX3*	+	4683–5471	789	ATG/TAA	8	4698–5486	789	ATG/TAA	9
*trnG* (Gly)	+	5480–5543	64	UCC	0	5496–5560	65	UCC	0
*ND3*	+	5544–5897	354	ATT/TAG	0	5561–5914	354	ATT/TAG	0
*trnA* (Ala)	+	5896–5962	67	UGC	8	5913–5977	65	UGC	−1
*trnR* (Arg)	+	5971–6034	64	UCG	3	5977–6039	63	UCG	2
*trnN* (Asn)	+	6038–6102	65	GUU	0	6042–6106	65	GUU	0
*trnS2* (Ser) (AGN)	+	6103–6170	68	GCU	0	6107–6173	67	GCU	0
*trnE* (Glu)	+	6171–6238	68	UUC	18	6174–6239	66	UUC	18
*trnF* (Phe)	−	6257–6322	66	GAA	0	6258–6324	67	GAA	0
*ND5*	−	6323–8042	1720	ATT/T-	15	6325–8044	1720	ATT/T-	15
*trnH* (His)	−	8058–8122	65	GUG	0	8060–8124	65	GUG	0
*ND4*	−	8123–9461	1339	ATG/T-	2	8125–9463	1339	ATG/T-	0
*ND4L*	−	9464–9754	291	ATG/TAA	2	9464–9754	291	ATG/TAA	2
*trnT* (Thr)	+	9757–9821	65	UGU	0	9757–9821	65	UGU	0
*trnP* (Pro)	−	9822–9887	66	UGG	3	9822–9887	66	UGG	2
*ND6*	+	9891–10,415	525	ATT/TAA	−1	9890–10,414	525	ATT/TAA	−1
*CYTB*	+	10,415–11,551	1137	ATG/TAG	0	10,414–11,550	1137	ATG/TAG	0
*trnS1* (Ser) (UCN)	+	11,550–11,616	67	UGA	15	11,549–11,615	67	UGA	18
*ND1*	−	11,632–12,568	937	ATA/T-	10	11,634–12,570	937	ATT/T-	10
*trnL2* (Leu) (CUN)	−	12,579–12,644	66	UAG	0	12,581–12,644	64	UAG	0
lrRNA	−	12,645–13,971	1327	/	0	12,645–13,964	1320	/	0
*trnV* (Val)	−	13,972–14,043	72	UAC	0	13,965–14,036	72	UAC	0
srRNA	−	14,044–14,827	784	/	0	14,037–14,824	788	/	0
D-loop region	+	14,828–16,499	1672	/	0	14,825–15,958	1134	/	0

## Data Availability

The raw sequencing data generated in this study have been submitted to the NCBI Short Read Archive (SRA, https://www.ncbi.nlm.nih.gov/sra, accessed on 6 September 2022) under accession numbers SRR21438445–55; the mitogenomes assembled in this study have been submitted to the NCBI under accession numbers OP381033–44.

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
