# Peer review of "Comparative Mitochondrial Genomes between the Genera Amiota and Phortica (Diptera: Drosophilidae) with Evolutionary Insights into D-Loop Sequence Variability"

_genes, 2023, doi:10.3390/genes14061240_

Round 1

Reviewer 1 Report

Dr Zhang and Colleagues have sequenced the complete mitochondrial genome in 11 steganine Drosophila species. Adding one genome from the literature, they analyze a dataset of 6 species from genus Amiota and 6 from Phortica. They provide an extensive description of the mitochondrial genomes, with a specific focus on the D-loop, and a phylogeny.

I think the study is well done and complete, and I would suggest the revision of some minor issues listed below.

The sign of ‘these authors contributed equally…’ points to one single author, please correct.

Abstract, final lines: I am not sure that ‘high evolutionary rates and genetic divergence’ in a few species is a sufficient basis for the establishment of new subgenera. Furthermore, raising new subgenera has significant outcomes in terms of the taxonomy of the group, especially in a key taxon as Drosophila, and should be carefully evaluated before making a formal statement. If a proposal is made, it should be well justified by different lines of evidence and analyses that target a vast number of species in the subgenera.

Page 2, line 9: ‘comprise’ here is not the correct term.

Page 2, line 42-43: it would be interesting to know, based on the current taxonomy and current knowledge on the phylogeny and diversity of the group, if the species sampling under scrutiny can be regarded as representative of the two genera or, else, other species groups within the genus are present and have not been included.

Page 3, 3 lines from the end: please include the accession number for the D. melanogaster genome.

Page 4 lines 6-7: ‘searched’ is not the correct term here.

Page 4 line 26: please add a clear statement that all sequenced genomes are complete or indicate where they are not complete.

Page 5 line 17: would the authors say that the difference is in the linear relationships between skews or in the fact that the two genera show non overlapping ranges in the skew values (see panel G).

Page 6 line 46: here I read a clear statement about size and AT content being correlated. If the basis for this statement is simply that Amiota sequences are long and A-T rich and sequences form Phortica are short and less A-T rich, this is a single observation and should not be reported as a correlation. If a proper correlation analysis is to be made, I assume there would be a problem with normality if the 12 sequences are joined together.

Paragraph 3.4: while I am not very familiar with the Drosophila literature, I would guess that a lot is known about the functioning of the Drosophila D-loop, including the presence of conserved elements and their role. If possible, it would be interesting to compare the motifs identified here with motifs known from other Drosophila species and their function.

Page 8 line 2-3: this sentence should be revised. I agree that if two markers evolve under severely different evolutionary pressures, the two markers may produce different phylogenetic trees, even if they share the same phylogenetic history. I am not sure the opposite is true. The observation that two markers recover the same phylogenetic history is simply what we expect if they both are capable and informative enough to reconstruct the correct (shared) phylogenetic history.

Figure 4 is difficult to read. Using a red/green scale for the heatmap may help.

Figure 7 is difficult to read at this scale, support values are not readable at all.

Some minor corrections are needed, but the language is generally appropriate.

Reviewer 2 Report

Dear Authors,

The manuscript is interesting but there are a lot of points that need your attention. Please find my comments in the attached file.

Best regards

Dear Authors,

English language is good.

Best regards

Round 2

Reviewer 2 Report

Dear Authors,

Thank you for the changes. I still think that you should write the Order and the family of each species. Other than that I think this paper should be published.

Best regards

Dear Authors,

English language is great!

Best regards